# What Is Heat? Can Heat Capacities Be Negative?

**DOI:** 10.3390/e25030530

**Published:** 2023-03-19

**Authors:** Emil Roduner

**Affiliations:** 1Institute of Physical Chemistry, University of Stuttgart, 70569 Stuttgart, Germany; e.roduner@ipc.uni-stuttgart.de; Tel.: +41-44-422-34-28; 2Department of Chemistry, University of Pretoria, Pretoria 0002, South Africa

**Keywords:** heat, work, deficiencies of bulk thermodynamics, negative heat capacities, entropy of self-gravitating systems, virial theorem and heat

## Abstract

In the absence of work, the exchange of heat of a sample of matter corresponds to the change of its internal energy, given by the kinetic energy of random translational motion of all its constituent atoms or molecules relative to the center of mass of the sample, plus the excitation of quantum states, such as vibration and rotation, and the energy of electrons in excess to their ground state. If the sample of matter is equilibrated it is described by Boltzmann’s statistical thermodynamics and characterized by a temperature *T*. Monotonic motion such as that of the stars of an expanding universe is work against gravity and represents the exchange of kinetic and potential energy, as described by the virial theorem, but not an exchange of heat. Heat and work are two distinct properties of thermodynamic systems. Temperature is defined for the radiative cosmic background and for individual stars, but for the ensemble of moving stars neither temperature, nor pressure, nor heat capacities are properly defined, and the application of thermodynamics is, therefore, not advised. For equilibrated atomic nanoclusters, in contrast, one may talk about negative heat capacities when kinetic energy is transformed into potential energy of expanding bonds.

## 1. Introduction

Heat is normally said to represent kinetic energy of atoms and molecules. This is not wrong, but it is incomplete as a definition. Joule, in 1843, showed that one can heat up matter that is in contact with a coil through which electricity is passed, and he concluded that heat is a state of vibration, not a substance [1]. More than a century later, the nature of heat was still questioned by Dyson, and the answer that he gave was “heat is disordered energy” [2]. His definition of heat still requires some more precision. He may have meant or should have said “heat is the energy of pure random motion under constant volume, excluding monotonic motion”. Expansion or contraction of a system against an external pressure involves volume work, which represents an amount of uniform motion. Pure random motion cannot do any work. Clausius, in his original work [3], described heat as “stationary motion”, carefully excluding “velocities that alter continuously in the same direction”, although he also included periodic motion such as “those of the planets about the sun”, which we believe is not correct.

In their work *Which physical quantity deserves the name “Quantity of heat”?,* Herrmann and Pohlig [4] suggested that this is the entropy *S.* This proposal disregards that *S* does not have the dimension of an energy. Furthermore, the change of internal energy dU of a system is equal to the exchange of heat δQ plus the work δW that is executed on the system (positive sign of δW) or by the system (negative sign), as given by the first law of thermodynamics:(1)dU=δQ+δW

Kinetic and potential energy are part of the internal energy and are state functions, but heat is not a state function, so there must be a difference between the two properties, and we can only equate heat with kinetic energy in the absence of work. It is essential that work is not mixed up with heat [5]. Most commonly in thermodynamics, δW is volume work, but there are many other forms, such as electrical or magnetic work, or gravitational work which is most relevant in the present context. In a similar sense, but on a very different length scale, there is bond dissociation work or work against non-covalent bonds in molecular associations.

We, thus, prefer the following more precise definition of heat: For a stationary system, i.e., in the absence of any work, the exchange of heat of a sample of matter equals the change of its internal energy, which is given by the kinetic energy of random translational motion of all its constituent atoms relative to their center of mass, plus the excitation of vibration and rotation and the energy of electrons in excess to their ground state. If the sample of matter is equilibrated, it is described by Boltzmann’s statistical thermodynamics and characterized by a temperature T.

While heat is not a state function, the reversibly exchanged heat divided by the temperature at which this exchange occurs is a state function and called entropy S, as concluded from the Carnot cycle [6]. For volume work, δW=−pdV:(2)$$dS\geq \frac{\delta Q_{rev}}{T}=\frac{dU+pdV}{T}$$

The equal signs hold for reversibly exchanged heat, i.e., for processes where pressure and temperature of a system and its environment are equal. This is the case in equilibrium where there is no net heat flow and the rate of entropy production in the process is zero. A spontaneous process with positive entropy production, dS>0, occurs only when there is a temperature gradient. The amount of heat Q flows only from hot to cold, which is one of the ways to express the second law of thermodynamics that defines the direction of spontaneous processes, the arrow of time, based on the positive sign of dS, the total entropy of the system plus the environment, often also called the universe. In the literature, thermodynamics is also applied to the real universe [7], which is problematic, as discussed below.

The entropy of an ideal gas depends mostly on temperature and on volume or concentration. For a one-component gas phase sample in equilibrium we have from Equation (2)
(3)$$TdS={\left(\frac{\partial U}{\partial T}\right)}_{V}dT+{\left(\frac{\partial U}{\partial V}\right)}_{T}dV+pdV=C_{V}dT+{\left(\frac{\partial U}{\partial V}\right)}_{T}dV+pdV$$

The internal energy U of an ideal gas is independent of volume. When it expands freely and, therefore, irreversibly from the initial volume $V_{i}$ to the final volume $V_{f}$, it does not do any work, the temperature, therefore, stays constant. For the integration of Equation (3), it is convenient to replace the irreversible path by an isothermal reversible path from the initial to the final state. The first two terms do not contribute to the change of entropy. Only the volume work term contributes, and with p=nRT/V and the concentration *c* = *n*/*V*, we obtain for the entropy change on free expansion.
(4)$$\Delta S=\int_{V_{i}}^{V_{f}}\frac{nR}{V}dV=nR\ln\left(\frac{V_{f}}{V_{i}}\right)=nR\ln\left(\frac{c_{i}}{c_{f}}\right)$$

This shows that the entropy of a freely expanding ideal gas increases logarithmically with volume or decreases with concentration, while its temperature remains constant.

We may worry whether temperature is still defined for a gas that expands into vacuum and arrives quickly at near collisionless concentrations, where the motion of the atoms is essentially radial and uniform. However, in the absence of collisions and of interatomic forces, the original random velocity vectors are conserved and the Maxwell-Boltzmann distribution that defines the temperature is retained.

## 2. Deficiencies of Bulk Thermodynamics

Thermodynamics is, in itself, a consistent theoretical formalism that describes properties of matter in a bulk view, as opposed to an atomistic view. All laws of thermodynamics are empirical, not derived from first principles. The problems arise when we apply the theory to matter. It quickly becomes apparent that the bulk continuum view of the classical thermodynamic properties needs refinement by inclusion of an atomistic picture.

The classical picture works rather well for non-interacting particles, such as ideal gases, for which it is obviously designed, and also for condensed matter, where the interactions are close to constant, apart from fluctuations due to fast random or periodic motion in which the interactions are averaged out. Deviations appear mostly in concentration ranges where particles pass through a distance range of typically very few atomic or molecular diameters, during which the interatomic Lennard–Jones-type potentials exceed or become competitive with thermal energy, and the distances undergo a net change (Figure 1). They are essential for the phase transitions of the condensation of gases and the solidification of liquids. For example, the entropy of vaporization of a liquid at its natural boiling point is normally calculated by dividing the latent heat of vaporization by the absolute boiling temperature. This does not represent the correct entropy difference between liquid and gas, since rather than representing the increased kinetic energy, part of the applied heat is used for the work of separating the associated molecules along the red arrow in Figure 1. Evidence for this is that the evaporation entropy of hydrogen bonded molecules for water (109 J/mol K) and methanol (104 J/mol K) is considerably larger than the common value of 88 J/mol K that is given by Trouton’s rule.

In physical chemistry, deviations from ideal behavior with a compression factor $Z=pV_{m}/RT=1$ are routinely accounted for by empirical correction factors. These fudge factors are normally relatively small, which has the advantage that one can often get an approximate picture assuming ideal behavior. If a higher accuracy is desired, one must include these corrections. There is a multitude of correction factors, such as activity coefficients, fugacities, virial and van der Waals coefficients, and Joule–Thomson coefficients.

Figure 1 illustrates that at low pressures or large average intermolecular distance all molecules behave as ideal gases. At much higher pressure, they reach the attractive distance range r<σ and, therefore, show 0<Z<1. Only at very high pressures, Z>1 is reached at repulsive average distances. H_2_ obviously shows near-negligible attraction.

For gases, Z<0, corresponding to absolute pressure <0, is not quite reached, but the van der Waals equation accounts for 0<Z<1 by the negative internal pressure (or cohesive pressure) correction term. On condensation, the negative internal pressure changes continuously into a negative absolute pressure. The latter has been discussed for complex liquids and solids with sufficiently high cohesive energies [8]. In conventional thermodynamics, a negative pressure state means that an increase of volume results in a decrease of entropy, warning that this is uniquely defined only for cases where either there is no work or no heat exchange performed during a process, and if heat and work appear simultaneously, only the internal energy can be measured [9]. This is in accord with our above recommendation for the definition of heat.

The bridge between the bulk and the atomistic pictures has been developed by Boltzmann in his statistical thermodynamics [10]. This has greatly advanced the understanding of the traditional thermodynamic properties. For example, it has been shown that the second virial coefficient B describes the deviation of the compression factor of a real gas from its value of Z=1 for the ideal gas. It relates to the two-body interatomic potential energy function, and it can be computed from statistical mechanics [11]. The Debye–Hückel limiting law accounts for concentration dependent activity coefficients of salt solutions, based on ionic strength, and related to the 1/r scaling of the Coulomb interaction. Salt solutions are always electrically neutral, although a given ion is preferentially surrounded by oppositely charged ions. Mixing ions of opposite charges leads to *effective shielding*, thus reducing the long-range tail of the Coulomb interaction significantly.

In contrast, celestial bodies, such as stars, suns, and galaxies, interact by the *unshielded* self-gravitational potential interaction term, $U_{g}=-Gm_{1}m_{2}/r$ (G = gravitational constant, $m_{1},\;m_{2}$ = masses). Even though the gravitation of, for example, an electron and a proton, is much smaller than the Coulomb interaction in a hydrogen atom and, therefore, negligible, gravitation becomes important at very large distances because it is proportional to the masses of both interacting bodies, and these are greater than those of a proton and electron by many orders of magnitude, while the amount of charges on these bodies and, therefore, their Coulomb interaction remains very low. Although gravitation represents an interaction that makes galaxies and the entire universe deviate much from an ensemble of non-interacting ideal gas atoms, thermodynamics is applied to the universe, and the second law of thermodynamics, the arrow of time, is used to predict whether the universe should expand or contract in future. Indeed, an inappropriate definition of heat has led to a phenomenon called the gravo-thermal catastrophe, suggesting negative heat capacities and non-monotonic behavior of entropy with the extent of expansion of the universe (see below) [12,13,14].

Kinetic energy of random motion can already be seen as an amount of heat. However, in most cases, and in particular for thermodynamic applications, one may want to understand heat as a property of a sample of matter in thermal equilibrium that can be characterized by a temperature. The kinetic energy of an equilibrated ideal gas is represented by a Maxwell–Boltzmann velocity distribution that fulfills the equipartition theorem, $E=\frac{1}{2}\;k_{B}T$ per fully excited degree of freedom. Additionally, a general sample of equilibrated matter must obey Boltzmann’s statistical mechanics. This represents a constraint in the definition of heat that is necessary if one wants to derive entropy, and for the application of thermodynamics in general.

A uniform potential executes a uniform force on a particle, which leads to the conversion of potential energy V into kinetic energy T of uniform motion. It is treated by the *virial theorem*, according to which V=−2T and the total energy E=−T for $-r^{-1\;}$ scaling of the potentials. This type of kinetic energy is directed, not random. It performs work, and it is essential that it is not mixed up with heat.

The kinetic energy of core electrons of an atom is much higher than that of its valence electrons. Nobody would say that this higher kinetic energy represents a higher temperature of the core electrons. According to the time-dependent Schrödinger equation, electron motion is not random, it is deterministic. In its orbit about the nucleus, it continuously interchanges part of its kinetic and potential energy, as described by the virial theorem.

## 3. The Self-Gravitating System of the Universe and Atomic Nanoclusters: Do Negative Heat Capacities Exist?

Celestial bodies interact gravitationally. We have already seen that matter moving under the action of a potential requires corrections to thermodynamic properties, which for chemical systems, depending on concentration, are in most cases factors below 3, and the range of the interacting potential is only a few atomic diameters.

Galaxies, and in fact the entire universe, are being treated by thermodynamics [7]. Of interest are the questions of their evolution by spontaneous expansion or contraction and of their temperature based on the second law of thermodynamics. This assumes that thermodynamics is a valid basis for the description of the universe, which is a tricky problem that is controversially discussed in the literature [15]. Expansion implies that the initial entropy near the Big Bang is low, which is fine for a small volume that can thereafter expand. On the other hand, the Big Bang requires an initially hot and, therefore, high entropy plasma. It is a question how the two entropy contributions during expansion and cooling are balanced. The issue is further greatly complicated by the fact that there are various other contributions to the entropy of the universe. By far the most important one being the black holes, followed by contributions of the cosmic microwave background, the interstellar and intergalactic medium, and only in seventh position the contribution of all stars, being 24 orders of magnitude less than that of the black holes [7]. It is unclear to what extent these contributions are coupled or whether they evolve mostly as non-equilibrated separate processes. Furthermore, on a length scale below a million light years, the universe is very inhomogeneous, which represents another complication for the application of thermodynamics. We shall restrict our discussion here to the galaxies and stars which at least resemble the kind of matter that is familiar to us in the laboratory, and we simply explore the effect of gravitation on a homogeneous fraction of the universe.

A sample of an ideal gas in equilibrium is characterized by the Maxwell–Boltzmann velocity distribution and is determined by the mass and temperature. The equilibrium is a result of multiple collisions. Secondly, such a sample acts as a Planckian black body that emits and absorbs photons which, in an isolated system at a given temperature, also leads to an equilibrium. The recoil of absorption and emission interferes with the particle velocities of the Maxwell–Boltzmann distribution so that the collisional and the radiative process are coupled. There is in fact only a single equilibrium and a single temperature [16]. This is different for the universe which is an ensemble of a large number of stars and galaxies. It is well known that the cosmic microwave background radiation obeys very well the wavelength distribution of a black body at ca. 2.7 K equilibrium temperature. Many of the celestial bodies are bright emitters of light with a wavelength distribution that also corresponds to a black body, but at a much higher temperature, such as 5778 K for our sun, suggesting that they can be treated by conventional thermodynamics at or near equilibrium. However, the velocity distribution of an ensemble of stars or galaxies is not determined by Maxwell–Boltzmann but is density dependent, distorted by gravity, and the recoil of microwave or visible light quanta on the velocities of the heavy masses is negligible. As shown below, such an ensemble is not an equilibrated system, and it cannot, in the same sense, be characterized by a temperature.

It is useful to first specify the system. We regard the universe as an isolated system consisting of an ensemble of N gravitational bodies. Since its volume must be flexible, the isenthalpic ensemble, in which H=U+pV is a conserved quantity, is the proper system [17]. For isenthalpic conditions:(5)$$dH={\left(\frac{\partial H}{\partial p}\right)}_{T}dp+{\left(\frac{\partial H}{\partial T}\right)}_{p}dT=0$$

The Joule–Thomson coefficient μ, given by the change in temperature with pressure
(6)$$\mu ={\left(\frac{\partial T}{\partial p}\right)}_{H}=-\frac{{\left(\frac{\partial H}{\partial p}\right)}_{T}}{{\left(\frac{\partial H}{\partial T}\right)}_{p}}=-\frac{\varepsilon }{C_{p}}$$
is used to describe the Joule–Thomson effect where, below an inversion temperature, a real gas cools on expansion even though the heat capacity $C_{p}$ is invariably positive and there is no heat exchange with an environment. ε describes the pressure dependence of the enthalpy across the expansion frit. Cooling occurs because the system performs work against the attractive (longer range) part of the intermolecular forces (Figure 1), and because of the many collisions there is a defined temperature. By analogy, the kinetic energy of the monotonic expansion of the universe is converted to potential energy in the gravitational field. There is no exchange of heat, but the system performs work against gravitation. In contrast to the real gas, collisions are absent and there is no defined temperature.

Similar effects occur near melting transitions of small, isolated clusters when controlled amounts of energy are added by irradiation with IR photons, as discussed for $Na_{147}^{+}$ clusters [18,19]. While adding energy to bulk systems causes progressive melting, nanoclusters try to avoid a partly molten state and prefer to convert a fraction of the added energy to potential energy by extending some of its bonds in a rearranged structure. By this internal energy interconversion, the cluster becomes colder, while its total energy increases, reflecting a negative heat capacity and giving rise to a “back-bending” in the entropy curve [18,19].

Negative heat capacities in nanoclusters have been suggested and debated under various experimental conditions [20,21,22]. Overall, there is no doubt, that the phenomenon exists. Rather than representing a special state of matter it occurs when a system converts some of its kinetic energy to potential energy in bonds. This often happens in inhomogeneous systems [19], when one part cools down while another part rearranges to a state of higher energy by expansion along a Lennard–Jones-type potential curve [23]. Care has to be taken for non-equilibrium evaporative ensembles in vacuum where neither the volume nor the applied pressure is defined [24], and for the calculation of entropies where δQ but not δW plays a role. Non-interacting ideal-gas-like bodies in an equilibrated state are characterized by a Maxwell–Boltzmann velocity distribution that defines a temperature of the system. For ions in solution, negative ions tend to accumulate near positive ions, and vice versa, which distorts the spatial distribution of particles towards clustering. In a similar sense, gravitation attracts and accelerates the particles at lower distances more than the remote particles. We, therefore, expect a larger contribution from low distances and, thus, a non-homogeneous density distribution. On contraction, the dominance of low distances and their kinetic energy increases, distorting the velocity distribution so that it deviates from the Maxwell–Boltzmann distribution [25]. Therefore, the classical definition of temperature as a precondition for the validity of much of thermodynamics no longer applies. In addition, pressure or heat capacity are not defined properties of the universe, and Equation (6) is not applicable. A negative heat capacity, as claimed in the gravo-thermal catastrophe [13], is a consequence of falsely interpreting the change of monotonic kinetic energy as heat instead of gravitational work [16]. The problem is related to the one discussed in Figure 1 for bond dissociation work, except that gravity scales as $-r^{-1}$ and is, therefore, of much longer range than the dissociative branch of the Lennard Jones potential that scales typically as $-r^{-6}$.

Nevertheless, Antonov and Lynden-Bell assumed the validity of thermodynamics for self-gravitating systems [12,13,14]. The phenomenon was called the “gravo-thermal catastrophe”. The astrophysical arguments on negative heat capacities all depend on the virial theorem [26], which describes the conversion between potential energy and kinetic energy of monotonic, not random, motion. These systems perform gravitational work, but heat is not even defined for a non-equilibrium ensemble of celestial bodies. Interpreting a decrease of kinetic energy in an expanding universe as a negative heat and deriving from it a change of entropy for a judgement of spontaneity in the second law of thermodynamics is not sound. It is no surprise that this leads to a strange behavior of the evolution of entropy [12,14].

He and Kang suggested a different way of escaping the dilemma. They proposed to generalize the second law of thermodynamics, according to which entropy never spontaneously decreases, by adding that “for the long-range interaction of self-gravitating systems the entropy never increases” [27]. This acknowledges the importance of long-range interactions which are not accounted for in conventional thermodynamics and amends the empirical law by an empirical addition. However, it does not account for the improper definition of heat in the presence of the long-range potential.

Attempted solutions to the debate have become available which use various approximate approaches up to the mean field approximation (which essentially represents a continuum) using classical and statistical thermodynamics [28]. However, modern cosmological models involve large amounts of dark energy to explain why the universe is actually accelerating, contrary to conventional believe [29]. It is generally believed that a gravitationally collapsing body will give rise to a black hole, and it was nevertheless suggested that a close analogy to entropy and temperature exists for its description. Four laws were formulated which “correspond to and in some way transcend the four laws of thermodynamics” [30].

What does this debate now mean for the cosmological picture of the evolution of the universe? Conventional thermodynamics predicts contraction as the spontaneous process. However, we know from the work of Hubble in the 1920s that the universe is expanding, a fact that is described by the Hubble constant, which is, of course, possible if expansion is not a spontaneous process but one that is driven. It is a relict initiated by the blast of the Big Bang, taking it to a high speed of expansion and to a low entropy state. This process is still going on, but it is continuously retarded by gravity and will stop and reverse to a contraction when the complete initial kinetic energy is used up and converted to potential energy. This is the Big Crunch model [29]. Furthermore, the −1/r scaling of gravity predicts that the Hubble constant is density dependent [31]. The position where the Hubble constant is measured will, therefore, contribute to its value, which is currently significantly disputed. Indeed, the farther away the galaxies are from the Milky Way, the faster they are receding.

## 4. Conclusions

It is essential to distinguish between heat and work. It is only in the absence of work that heat is equivalent to the kinetic energy of random motion plus the excitation energy of quantum states in a sample of matter. Equilibrated samples are characterized by a temperature. The addition of heat to equilibrated nanoclusters near a phase transition may lead to expansion of some of the cluster bonds, performing work along the attractive potential energy curve of the bond, under simultaneous cooling of another part of the cluster, mimicking a negative heat capacity. However, kinetic energy of monotonic (collisionless) motion and its interconversion with potential energy represents work but is not part of heat. The virial theorem that relates the conversion between kinetic and potential energy should, therefore, not be used in context with heat.

The cosmic microwave background indicates a Planck black body radiation that represents an equilibrium origin at a temperature of 2.7 K. In the same sense, light emission from stars represents a body in thermal equilibrium, although at much higher temperature. The monotonic motion of stars and galaxies, however, occurs in a self-gravitational potential and is not equilibrated. The interpretation of its kinetic energy as heat leads to false conclusions such as negative heat capacities. In the absence of collisions, temperature and heat capacity of an expanding or contracting universe are not defined. For this reason, and because of the not yet understood dark energy, classical thermodynamics is not applicable to the universe.

## Figures and Tables

**Figure 1 entropy-25-00530-f001:**
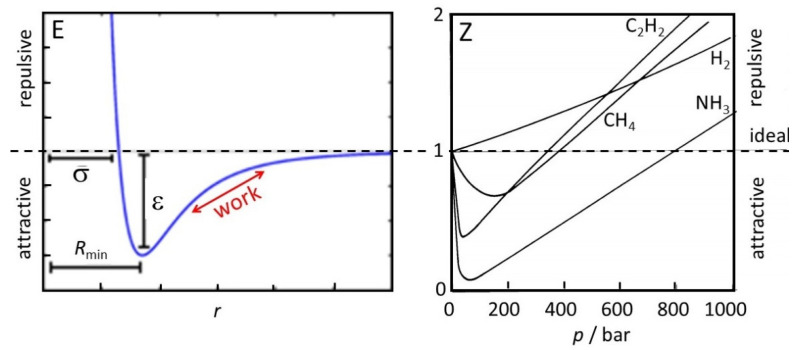
Schematic view of Lennard–Jones-type intermolecular potential energy E as a function of distance r (**left**), and compression factor Z for some representative molecules at room temperature (**right**). The broken line represents the rigid sphere potential for non-interacting ideal gas atoms, corresponding to Z=1. Below this line is the effect for the attractive range of the potential, above the line that for repulsion.

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
