# Peer review of "What Is Heat? Can Heat Capacities Be Negative?"

_entropy, 2023, doi:10.3390/e25030530_

Round 1
Reviewer 1 Report
Report of “What is heat - can heat capacities be negative?” by Emil Roduner, submitted for Entropy
I started to read this manuscript with a lot of interest. It is well-written and interesting up to line 261, but it abruptly ends. In the title, it is promised to tackle the question of the existence of negative heat capacity, but at the end (although there is some discussion about this quantity), the conclusion is about the general invalidity of the second law (or rather, the present “classical thermodynamics”) on the Universe. I think t is not really a novel result, the Universe is too complex to be described by classical AND equilibrium thermodynamics.
In the present form, the manuscript is not complete, and I would recommend the Author tackle a bit more – as promised in the title - the question of negative heat capacity. Even in “normal” systems, heat negative heat capacity can be seen in non-equilibrium systems for describing instable states (see for example Lukács, B., Martinás, K. (1990) Thermodynamics of Negative Absolute Pressures, Acta Phys. Pol., B21, 177-183; Debenedetti, P.G.Metastable Liquids: Concepts and Principles; Princeton University Press: Princeton, NJ, USA, 1996; Imre,A., Martinás,K., Rebelo, L.P.N.: Thermodynamics of Negative Pressures in Liquid. J. Non-Equilib. Thermodyn. 1998, 23, 351–375; A. R. Imre, K. W. Wojciechowski, G. Györke, A. Groniewsky and J. W. Narojczyk: Pressure-volume work for metastable liquid and solid at zero pressure, Entropy 20 (2018) 338; doi:10.3390/e20050338). The Universe is rather a non-equilibrium than a classical system, therefore, the existence of negative heat capacity ( and other anomalous response functions, like negative compressibility) is inevitable.
My recommendation is “Major Revision”.
Reviewer 2 Report
See attached report

Round 2
Reviewer 1 Report
Report of the revised version of “What is heat - can heat capacities be negative?” by Emil Roduner, submitted for Entropy
The Author accepted my comments and changed the manuscript according to my recommendations. Some part is still unclear to me, but it is probably caused by the lack of my improper knowledge in thermodynamics.
I think the manuscript is interesting and probably will induce intense discussions within the field.
My recommendation is to accept it in its present form.
Reviewer 2 Report
As the author explains in his reply, this contribution is indeed an opinion, commissioned by a guest editor. Being this the case and, in addition, having the author corrected the minor inaccuracies that were pointed out, as well as including further discussion, to make the point of the paper, I have no further observations. Therefore, I can recommend the article for publication.